# Primary Sinonasal Mucosal Melanoma: A Narrative Review

**DOI:** 10.3390/diagnostics15040496

**Published:** 2025-02-18

**Authors:** Nikola Todorovic, Petar Djurkovic, Aleksandar Krstic, Nada Tomanovic, Pavle Milanovic, Djurdjina Kablar, Zlata Rajkovic Pavlovic, Momir Stevanovic, Jovana Milanovic, Aleksandra Arnaut, Ivan Ljubisavljevic, Dragica Selakovic, Gvozden Rosic, Milica Vasiljevic

**Affiliations:** 1Clinic for Otorhinolaryngology and Maxillofacial Surgery, University Clinical Center of Serbia, 2 Pasterova Street, 11000 Belgrade, Serbia; nikola.todorovic89@hotmail.com (N.T.); petardjurkovic92@gmail.com (P.D.); krstichi@gmail.com (A.K.); 2Faculty of Medicine, University of Belgrade, 11000 Belgrade, Serbia; nada.tomanovic@med.bg.ac.rs; 3Institute of Pathology, 1 Dr. Subotica Street, 11000 Belgrade, Serbia; 4Department of Dentistry, Faculty of Medical Sciences, University of Kragujevac, 34000 Kragujevac, Serbia; pavle11@yahoo.com (P.M.); zlatakg@yahoo.com (Z.R.P.); momirstevanovic7@gmail.com (M.S.); jovannakg94@gmail.com (J.M.); sandra11_92@yahoo.com (A.A.); ivanlj@gmail.com (I.L.); 5Department for Pathology, Pathohistology and Medical Cytology, University Clinical Centre of Serbia, 11000 Belgrade, Serbia; kablar.djurdjina@gmail.com; 6Department of Physiology, Faculty of Medical Sciences, University of Kragujevac, 34000 Kragujevac, Serbia; dragica984@gmail.com (D.S.); grosic@medf.kg.ac.rs (G.R.)

**Keywords:** mucosal melanoma, sinonasal melanoma, maxillectomy, prognosis

## Abstract

Mucosal melanomas (MMs) are under-researched tumors with a poor prognosis that arise from melanocytes found in the mucous membranes at different anatomical locations throughout the body. MMs are an uncommon yet highly aggressive tumor that typically develops on mucosal surfaces, predominantly in the head and neck region. MM of the head and neck occurs in 8–10% of all head and neck melanomas. It most commonly affects the mucosa of the nasal cavity and paranasal sinuses (75%), followed by the oral cavity mucosa (25%). A family history and the presence of mucosal nevi are associated with the occurrence of MM. Inhaled and dietary carcinogens are also linked to the development of sinonasal melanoma, much like other malignancies of the nasal cavity. Overexpression of the *C-KIT* gene is found in more than 80% of all primary mucosal melanomas, with somatic mutations in 10–30% of cases. The presence of these genetic alterations is also reflected in recent clinical studies of specific gene inhibitors that have proven efficiency in the systemic therapy of melanoma. There are various treatment modalities for MM. Surgical therapy involves en bloc surgical resection with a macroscopically visible and palpable mucosal margin of 1.5–2 cm. Partial resection of the maxilla may be considered if it ensures adequate tumor-free margine. Because of its rarity, outcome data for locally advanced head and neck MM is limited and primarily derived from retrospective studies with small case numbers. This review aims to update and summarize findings from clinical trials, prospective observational studies, and retrospective studies, while also exploring future directions for multimodal treatment approaches in this area.

## 1. Introduction

Mucosal melanomas (MMs), first described in 1856, are under-researched tumors that arise from melanocytes within the mucosal membranes of the nasal cavity, sinuses, oral cavity, lips, pharynx, vulva, vagina, uterus, anorectal region, and other mucosal surfaces [1]. The average age at diagnosis for MM is approximately 70 years, compared to cutaneous melanoma, where the median age is 55 years [2,3]. MM of the head and neck occurs in 8–10% of all head and neck melanomas. It most commonly affects the mucosa of the nasal cavity and paranasal sinuses (75%), followed by the oral cavity mucosa (25%) [4,5]. Primary sinonasal mucosal melanoma is a rare and highly aggressive oncological entity [6]. The etiological factors contributing to mucosal melanoma and its pathogenesis remain unclear, and no definitive risk factors have been identified for its development so far [7,8,9,10,11]. Symptoms of sinonasal melanoma commonly include epistaxis, unilateral nasal obstruction and visible masses, which in some cases lead to facial morphological changes, pain, hyposmia, frontal headache, diplopia, epiphora and, ptosis [12,13]. Due to their concealed location and absence of noticeable early symptoms, MMs are often diagnosed at an advanced stage [14]. Sinonasal melanoma has a high propensity for both regional and distant metastases. Regional spread often involves cervical lymph nodes, while distant metastases commonly affect the lungs, liver, and other sites [15,16]. Despite the absence of a universally agreed-upon treatment protocol, wide surgical excision remains the primary approach for managing these tumors [17,18,19]. Sinonasal melanoma is associated with a poor prognosis, with 5-year survival rates varying between 20% and 55% [20]. Tumor location plays a critical role in prognosis, with worse outcomes observed for patients with sinus involvement [21]. In recent years, sinonasal melanoma has been considered and managed as a separate disease from cutaneous melanoma, including its staging system and treatment modalities [22].

## 2. Materials and Methods

This narrative review was conducted using electronic scientific databases, including PubMed (Medline), Wiley Online Library, The Cochrane Library, Google Scholar, and EMBASE. The search was designed to identify relevant studies without time restrictions, including all articles available up to 10 November 2024. Studies were included if they were written in English, focused on human subjects, provided free full-text availability, and addressed the clinical characteristics, management, or outcomes of sinonasal mucosal melanoma. English-language-based materials such as reviews, online reports, original clinical studies, systematic reviews, and case reports were also considered.

Articles were excluded if they did not provide specific information on the clinical form of sinonasal mucosal melanoma, contained unverified or inconsistent data, or were duplicates identified during the review process. The search utilized key phrases, including “*Mucosal melanoma*”, “*Sinonasal melanoma*”, and “*Sinonasal mucosal melanoma*” to retrieve relevant studies. Titles and abstracts were screened, and eligible full-text articles were reviewed for relevance and data extraction.

Out of 531 initially identified studies, duplicates and articles failing to meet the inclusion criteria were removed. After a thorough screening process, 111 studies were included in the final analysis, ensuring extensive coverage of the epidemiology, etiopathogenesis, management, and prognosis of sinonasal mucosal melanoma. The process of selection is shown in the PRISMA flow diagram (Figure 1).

## 3. Discussion and Literature Review

### 3.1. Epidemiology, Etiopathogenesis, and Histopathological Features

Melanomas are aggressive tumors that originate from pigment-producing cells called melanocytes [23]. During embryonic development, melanocyte stem cells migrate from the neural crest to populate the skin and mucosal membranes [24,25]. Although predominantly arises in the skin, it can also develop in other extracutaneous sites containing pigment-producing cells [1]. Most melanocytes are found in the epidermis and dermis of the skin, but they are also present in other areas, including the eyes, mucosal membranes, and leptomeninges [26]. Mature melanocytes are elongated, dendritic cells located in the basal layer of the epidermis [27,28]. They are primarily responsible for the production of melanin, which serves several important functions in the body, such as protection from UV radiation contributing to skin pigmentation and antioxidant activity [29,30]. However, melanocytes are present in several sun-protected areas of the body, including mucosal membranes, where they are not required for sun protection [1]. The physiological functions of mucosal melanocytes are also not yet fully defined [25]. But in mucosal melanocytes, their immunological and antimicrobial functions are highlighted. Melanocytes have the ability to function as antigen-presenting cells, stimulate T-cell proliferation, and phagocytize microorganisms. As a result, melanocytes and their secreted products play a role in inhibiting the proliferation of bacterial and fungal pathogens [31,32].

Mucosal melanoma (MM) is a rare form of melanoma, accounting for roughly 1% of all diagnosed melanoma cases [33]. According to the American College of Surgeons, MMs represent a mere 1.3% of all melanoma cases identified over a nine-year period. Among these, the head and neck are the most frequently affected regions, accounting for over 50% of the cases [34]. In a study conducted by Colleen McLaughlin and colleagues [35], the paranasal sinuses are the second most common site for non-cutaneous melanomas, following anorectal melanomas, which have the highest incidence among these types. Furthermore, MMs contribute to approximately 9% of all malignant tumors found in the head and neck area [36]. Unlike cutaneous melanoma, which has a steadily increasing incidence, the occurrence of MM remains relatively stable (Table 1). It is also more commonly observed in women than in men, with an incidence rate of 2.8 per million compared to 1.5 per million in men, whereas cutaneous melanoma tends to occur slightly more often in men than in women [2,37]. A study by Lian and colleagues [38], involving 706 MM patients, found that 63% were women. This finding is consistent with those of Lund and coworkers [39] on sinonasal melanoma, as well as a 2022 study, which reported a male-to-female ratio of 1:1.1 for sinonasal melanoma cases [40]. However, most studies indicate that head and neck melanomas are equally distributed between genders, with some suggesting they are slightly more common in men [7,35,41]. These discrepancies may be attributed to factors such as sample size, geographic variations, and study design. Mihajlovic and colleagues [1] state that female predominance is mainly due to higher rates of genital tract melanomas, which account for 56.5% of MMs in women, while there is no gender difference in rates for extragenital MMs (Table 1). MM does not appear to show a strong racial preference overall. It is widely recognized that melanin pigmentation levels differ significantly across racial and ethnic groups and can also vary within individuals of the same racial or ethnic background. Melanin pigmentation is particularly common in the oral regions of Black individuals and is also more frequently observed in darker-skinned Caucasians compared to those with lighter skin tones [42,43,44]. McLaughlin and colleagues [35] report a significant racial difference in melanoma incidence, with a white-to-black ratio of 16:1 for cutaneous melanoma. However, for MMs in the head and neck region, the racial disparity is less pronounced, with a 2:1 ratio. While MM does not appear to have a clear racial preference, it is more common in Black, Asian, and Hispanic populations, likely due to the lower rates of cutaneous melanoma in these groups. Additionally, oral cavity MM is more frequent in Black and Japanese populations [7,35]. MM are more commonly diagnosed in older individuals compared to cutaneous melanomas, likely due to distinct biological characteristics and associated risk factors. MM of the head and neck is typically diagnosed between the ages of 60 and 69 [45,46], aligning with the findings on sinonasal melanoma reported by Roth et al. [47] and Temmermand et al. [48], where patients aged 65 and older accounted for the highest proportion of cases (Table 1). Shah and colleagues [49] reported a rare case of sinonasal melanoma in a 4-year-old patient, while Ravid and coworkers [50] documented the youngest known patient, an 8-month-old Black girl. However, oral cavity melanomas are often detected at a younger age, with up to 18% of cases found in individuals under 40 years old [45,46].

As mentioned, MM can occur on any mucosal surface, but they predominantly originate in the mucosa of the head and neck (55.4%), followed by the anus and rectum (23.8%), the female reproductive tract (18%), and the urinary tract mucosa (2.8%) [34,35,51]. Jethanamest et al. [52] analyzed 815 cases of head and neck MM from 1973 to 2007 using data from the National Cancer Institute. They identified the nasal cavity as the most frequently affected site (49.1%), followed by the paranasal sinuses (23.1%) and the oral cavity (18.8%). Lerner and collaborators [45] concluded that sinonasal melanomas predominantly originate in the nasal cavity, comprising 80% of cases. The most commonly affected areas are the turbinates and nasal wall, followed by the nasal septum. Among the sinuses, the maxillary sinus is the most frequently involved, with the ethmoid, frontal, and sphenoid sinuses being less commonly affected [45]. MM in the nasal cavity, paranasal sinuses, and nasopharynx is an uncommon disease, accounting for roughly 4% of sinonasal cancers. The reported incidence is 0.3 per million for nasal cavity cases and 0.2 per million for those involving the paranasal sinuses. Studies show that melanocytes are present in the sinonasal mucosa in approximately 21% of people [53,54]. Oral cavity melanomas typically affect the palate and the maxillary gingiva [55,56,57]. MMs of the head and neck, including sinonasal and oral types, may originate in areas of mucosal hyperpigmentation. Around 30% of oral MMs are linked to hyperpigmented sites, although the connection between these lesions and melanoma development remains uncertain [25,42,58]. Ascierto et al. [59] noted that MMs in the head and neck region present diagnostic challenges due to their multifocal nature (20% of cases) and the high prevalence of amelanotic forms (up to 40%), complicating early detection and treatment.

The causes and mechanisms behind MM remain largely unknown (Table 1), with no conclusive risk factors identified so far [7,8,9,10,11]. In contrast to cutaneous melanoma, the usual anatomical sites for MM are not exposed to UV light, which rules out UV exposure as a risk factor [8,60,61]. Multiple studies have found no link between exposure to carcinogenic viruses, such as human papillomaviruses, polyomaviruses, and human herpesviruses, and the etiopathogenesis of MMs [60,62,63]. In the sinonasal area, melanocytes appear to be involved in the metabolism of polycyclic aromatic hydrocarbons, suggesting a potential link between environmental inhalants, immune responses, and the development of sinonasal melanoma. This points to a possible environmental or immune factor contributing to its pathogenesis [7]. Exposure to formaldehyde has been identified as a possible risk factor for sinonasal mucosal melanoma, with reported cases of this rare malignancy in workers exposed to the substance in their professions. Likewise, cigarette smoking has been linked to oral MM, as oral pigmented lesions are more frequently observed in smokers. However, there is no conclusive evidence to suggest that common carcinogens like tobacco and formaldehyde play a role in the development of MM [7,64,65]. It is noted that genetic factors might contribute to the development of MM, as indicated by a family history of cutaneous melanoma [66]. MM and cutaneous melanoma differ significantly in their genetic mutation patterns, reflecting variations in their underlying causes and environmental influences. Although *BRAF* and *NRAS* mutations are predominantly found in cutaneous melanoma, where they account for approximately 75% of cases, they also play a significant role in the development of MM. MM frequently exhibits alterations in *KIT*, *TP53*, and signaling pathways such as *PI3K/AKT* and *MAPK*. *KIT* mutations, observed in 15–25% of MM cases, are notable due to their responsiveness to targeted therapies. Curtin and coworkers [67] noted overexpression of the *C-KIT* gene is found in more than 80% of all primary mucosal melanomas, with somatic mutations in 10–30% of cases. Moreover, MM often displays chromosomal abnormalities, including amplifications in *CDK4*, *CCND1*, and *TERT*, which contribute to its aggressive nature. *BRAF* and *NRAS* mutations, while prominent in cutaneous melanoma, are also detected in over 20% of MM cases, particularly in the head and neck mucosa [3,42,68,69,70,71].

Mucosal melanomas exhibit a variety of histological patterns, making them challenging to identify. These tumors may show spindle-shaped, epithelioid, or mixed cell morphologies, with spindle cells being elongated with eosinophilic cytoplasm, epithelioid cells being larger and polygonal with abundant cytoplasm, and mixed types containing both features. Sinonasal melanomas share common characteristics with other mucosal melanomas but have site-specific differences, including an infiltrative growth pattern, high mitotic activity, and frequent necrosis and ulceration. Vascular invasion is common, which contributes to their metastatic potential. Immunohistochemical staining, revealing markers like S-100, HMB-45, and Melan-A, plays a crucial role in diagnosis, while negative markers like cytokeratins help distinguish them from carcinomas. Differential diagnosis includes undifferentiated carcinomas, lymphomas, olfactory neuroblastomas, and sarcomas. Histologically, mucosal melanomas often lack pigmentation, complicating diagnosis, and may show high mitotic rates linked to worse outcomes. Given the wide range of morphological features, these melanomas can overlap with various sinonasal tumors, making immunohistochemical evaluation essential for accurate classification [53,67,72,73,74,75].

### 3.2. Diagnosis, Staging and Therapy

When diagnosing primary MM, especially in rare locations, it is essential to exclude metastatic lesions from cutaneous or ocular melanoma [1]. Diagnosis of sinonasal mucosal melanoma is often delayed due to the lack of distinct early symptoms and the difficulty in accessing lesions located in hard-to-reach areas during physical examination [41,76]. Amelanotic forms of MM further complicate the diagnosis [77]. The study by Lombardi et al. [78] investigated 58 cases of sinonasal mucosal melanoma over a 12-year period, revealing that the most common presenting symptoms included nasal obstruction (79.3%), epistaxis (70.7%), anosmia (25.9%), and headache (12.1%). Other less frequent symptoms included rhinorrhea (8.6%), epiphora (5.2%), and visual impairments, diplopia, unilateral facial swelling, and nasal pain (3.4% each). Interestingly, five patients (8.6%) were diagnosed incidentally without exhibiting any symptoms. However, Moreno and colleagues [79] reported that epistaxis is the most frequently observed presenting symptom in patients with sinonasal melanoma. On endoscopy, MM typically appears as a polypoid, unilateral lesion with varying pigmentation (brown, black, red, or pale white colors), sometimes accompanied by satellite lesions [1,13]. A biopsy is crucial for diagnosing sinonasal melanoma, as its histopathological features can vary. Immunohistochemical analysis, typically following hematoxylin-eosin staining, remains the gold standard for accurate diagnosis. Additionally, molecular testing for genetic mutations such as *NRAS* and *c-KIT* may assist in further characterizing the tumor [80,81]. A comprehensive history, physical examination, and imaging studies (including CT or MRI with contrast) are essential for assessing the extent of sinonasal malignant melanoma. In cases of sinus involvement, further imaging like chest CT, PET CT, or brain MRI may be needed to detect metastasis. CT scans typically reveal soft tissue opacification, often with or without bone erosion, while MRI is valuable for differentiating soft tissue masses and assessing skull base involvement. To evaluate regional and distant metastases, imaging of the neck, chest, and abdomen is required, with PET scans potentially offering higher sensitivity for detecting metastases [2,59,82].

The primary tumors in these anatomical regions are surrounded by extensive vascular and lymphatic networks, which facilitate the diffuse spread of the disease. Sinonasal melanoma has a strong tendency for both local and distant spread. It is reported that, at the time of diagnosis, cervical lymph node metastases are present in 10–20% of sinonasal melanoma cases, while distant hematogenous metastases, including those to the lungs, brain, bone, and liver, are detected in 6% of patients [79,83]. A study of 706 patients by Lian et al. [38] reported that 44% presented with metastatic disease—21.5% with regional spread and 23% with distant metastases. This is significantly higher than the 14% metastasis rate observed in cutaneous melanoma, highlighting the aggressive nature of MM. Despite this, it is noted that sinonasal melanomas are less likely to involve regional lymph nodes than MMs in other head and neck areas, reflecting differences in metastatic patterns [84].

There is no standard staging system for MMs. Various staging systems are used depending on the anatomical location, often borrowing from those applied to other common cancers of the same site. Accurate staging is essential after diagnosing malignancy, as it provides important prognostic information and helps guide treatment decisions [1,42,85]. MM of the aero-digestive tract is globally staged according to the American Joint Committee on Cancer (AJCC) system. The AJCC staging system excludes the T1 and T2 classifications, categorizing all tumors as T3 or T4 [42,86]. The prognostic grouping based on T, N, and M classifications was removed in the 8th edition of the AJCC staging [87]. Traditionally, primary MM of the head and neck has been staged using Ballantyne’s clinical staging system from 1970, which classifies the disease into three stages: stage I for localized disease, stage II for regional nodal involvement, and stage III for distant metastases [7,66,88]. Lian and coworkers [38] proposed a TNM staging system for MM, where T1 refers to a tumor that invades the submucosa, and T2 is used for tumors invading the muscularis propria. T3 is applied for tumors invading the adventitia, while T4 denotes tumors that invade adjacent structures. For regional metastasis, N1 is assigned to one affected lymph node, N2 for 2–3 nodes, and N3 for four or more nodes. M1 is used to indicate distant metastases. Unlike cutaneous melanoma, MMs currently lack standardized staging criteria based on tumor thickness or invasion depth. This absence of established thresholds makes it more challenging to categorize the progression of MMs [89]. There are other staging systems for MM, but they are not widely adopted [53,77,90].

Surgical excision remains the primary treatment for sinonasal melanoma, but achieving negative surgical margins can be challenging due to the complex anatomy of the sinonasal region, multifocal nature of the disease, and lentiginous growth patterns [39,42,87,88,91,92]. While neck dissection is commonly performed to address regional lymph node metastases, the routine use of elective neck treatment in the absence of clinically positive lymph nodes remains a subject of ongoing debate [93]. Radiotherapy can be an effective option, either as a primary treatment for unresectable or locally advanced MM or as an adjuvant therapy following surgery [94,95]. However, the role of chemotherapy and immunotherapy in improving overall survival for sinonasal melanoma has not been fully established through large-scale database studies [93]. Molecular profiling for mutations in genes such as *BRAF* and *KIT* is recommended, as it may help identify potential candidates for targeted therapies [45]. Although standardized treatment protocols for sinonasal melanoma are lacking, combining surgery with additional therapeutic strategies generally leads to improved outcomes, as single-modality treatments (either surgery alone or nonsurgical approaches) tend to be associated with poorer prognoses [96,97,98]. Treatment for sinonasal melanoma primarily focuses on surgery, with wide en bloc resection aiming for clear margins of at least 1.5 to 2 cm. When achieving adequate margins is challenging, partial resection of the mandible or maxilla may be necessary. Endoscopic techniques, which offer outcomes comparable to traditional approaches, are considered based on the likelihood of obtaining clear margins. For advanced tumors (Stage III, IVA), the National Comprehensive Cancer Network (NCCN) guidelines [4], reflecting the 2017 AJCC classification [87], recommend surgery with elective neck dissection and adjuvant radiotherapy to improve local control, though the impact on overall survival remains limited. For advanced-stage disease (Stage IVB, IVC), primary radiotherapy, systemic therapy, or clinical trial enrollment are often the preferred approaches. In cases where margin status is uncertain or anatomical complexities hinder margin assessment, adjuvant radiotherapy (at a dose of 54 Gy or higher with standard fractionation) may be indicated. Despite achieving clear margins and local control with R0 resections, sinonasal melanoma remains highly prone to local recurrence and distant metastasis, presenting significant treatment challenges [4,86,87,99]. MMs generally respond poorly to radiotherapy, limiting its use to unresectable or palliative cases [100]. Further research into the molecular distinctions between mucosal and cutaneous melanoma is essential for developing tailored treatment strategies. Immunotherapy and systemic therapies are under active investigation for their potential role in managing sinonasal melanoma [101,102,103].

### 3.3. Follow-Up Care and Prognosis

Follow-up care for sinonasal melanoma involves routine clinical examinations, including fiberoptic evaluations, especially when recurrence is suspected. During the first year, follow-ups are scheduled every three months, transitioning to every two to six months in the second year, every four to eight months between the third and fifth years, and annually thereafter. Radiological imaging of the head and neck is advised every six months to monitor the primary tumor site. Despite the complexity of managing MM, early detection and timely, aggressive treatment remain crucial for improving outcomes [98].

Patients with MM, including sinonasal melanoma, typically have a poorer life expectancy compared to those with cutaneous melanoma [42]. Abt and colleagues [104] reported that the survival rate for mucosal melanomas is approximately 25%, compared to 80% for cutaneous melanomas. Sinonasal melanoma often progresses aggressively, with outcomes significantly impacted by late diagnoses and a strong tendency for distant metastases [105]. These challenges arise due to the lack of early symptoms and the tumors’ location in hard-to-access areas, contributing to a high recurrence rate exceeding 50% and a 5-year survival rate of approximately 30.69% [106,107,108]. Additionally, the average survival time ranges from 17 to 28 months [13,87]. Tumors located in the nasal cavity generally have a better prognosis than those in the sinuses, likely due to earlier detection and easier access for treatment [22,39,109]. Dauer et al. [110] highlighted notable differences in survival based on tumor location, reporting a 3-year survival rate of 25.3% for melanomas in the maxillary sinus compared to 91% for septal melanomas without maxillary sinus involvement. The prognosis of sinonasal melanoma is influenced by factors such as tumor size, anatomical location, vascular invasion, mitotic activity, and cellular pleomorphism. Additionally, the presence of regional or distant metastases, advanced disease stage, older patient age, and tumors exceeding 3–4 cm are associated with less favorable outcomes [111].

## 4. Conclusions

In conclusion, sinonasal melanoma is a rare and aggressive malignancy with a poor prognosis, characterized by late-stage diagnoses and a high recurrence rate. This review, emphasizes the complexity of diagnosing and managing sinonasal melanoma. Despite the mainstay treatment of wide surgical excision, local recurrence and distant metastases, particularly to the lungs and lymph nodes, remain significant challenges. Multimodal approaches involving surgery, radiation, chemotherapy and immunotherapy are essential for improving outcomes, but their efficacy is limited by the disease’s aggressiveness and the difficulties in achieving clear surgical margins. The role of genetic factors, including *C-KIT* overexpression, highlights the potential for targeted therapies, although more research is needed to develop effective treatment strategies. Overall, early detection and personalized, aggressive treatment plans are critical in managing sinonasal melanoma, and continued investigation into its molecular and genetic profiles will be key to improving patient outcomes.

## Figures and Tables

**Figure 1 diagnostics-15-00496-f001:**
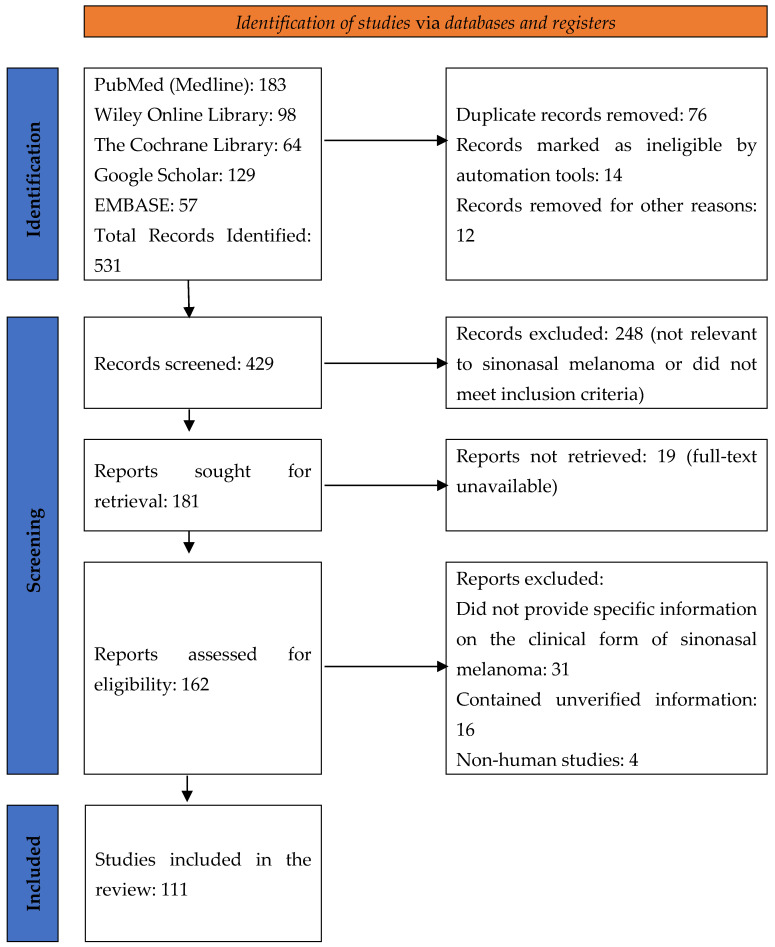
PRISMA flow diagram illustrating the study selection process.

**Table 1 diagnostics-15-00496-t001:** Comparison of Selected Characteristics of Cutaneous and Mucosal Melanomas.

Characteristic	Cutaneous Melanoma	Mucosal Melanoma
**Incidence**	More common, steadily increasing	Rare, accounts for ~1% of melanoma cases, stable
**Risk Factors**	UV exposure, fair skin, sunburn history	Unknown
**Common Sites**	Skin, particularly sun-exposed areas (e.g., back, legs)	Mucosal membranes (head and neck)
**Age Group Affected**	Typically diagnosed between 50–70 years	Primarily affects older individuals (~70 years)
**Gender Distribution**	Slightly more common in men	More common in women, but varies by location
**Racial/Ethnic Distribution**	More common in fair-skinned individuals, especially Caucasians	More common in Black, Asian, and Hispanic populations
**Genetic Mutations**	*BRAF*, *NRAS* mutations frequent, with *KIT* mutations in some cases	*KIT* mutations common (15–25%), *BRAF*, *NRAS* mutations also found
**Histology**	Often pigmented, can be amelanotic	Often pigmented, but amelanotic forms are more common
**Clinical Presentation**	New or changing mole, ulceration, bleeding	Nasal obstruction, epistaxis, anosmia, headache (for sinonasal melanoma)
**Diagnosis**	Skin biopsy, dermoscopy, sentinel lymph node biopsy	Endoscopy, biopsy, imaging (CT, MRI) for tumor extension
**Prognosis**	Generally better with early detection, 5-year survival ~90%	Poor prognosis, 5-year survival ~30%, high recurrence rate
**Metastasis**	Commonly to skin and regional lymph nodes and distant sites (lungs, liver, bone)	Local (lymph nodes) and distant metastases (lungs, brain, bones)
**Treatment**	Surgery, immunotherapy (*PD-1* inhibitors), targeted therapies, radiation	Primarily surgery with wide excision, adjuvant radiotherapy, chemotherapy in advanced cases
**Staging**	AJCC staging system with focus on tumor thickness and lymph node involvement	Less standardized staging, often uses AJCC or other clinical systems for tumor location
**Recurrence**	Risk of recurrence even after treatment, particularly in advanced stages	High recurrence rate, particularly in sinonasal melanoma

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
