# Peer review of "Primary Sinonasal Mucosal Melanoma: A Narrative Review"

_diagnostics, 2025, doi:10.3390/diagnostics15040496_

Round 1
Reviewer 1 Report
Comments and Suggestions for Authors
Todorovic et al. presented a narrative and review of primary sinonasal mucosal melanoma. However, they did not use the PRISMA Flow Diagram, which would have been desirable. Otherwise, the paper itself is interesting, systematically explains the topic, and with minor revisions could be interesting for publication.
Author Response
Comment: Todorovic et al. presented a narrative and review of primary sinonasal mucosal melanoma. However, they did not use the PRISMA Flow Diagram, which would have been desirable. Otherwise, the paper itself is interesting, systematically explains the topic, and with minor revisions could be interesting for publication.
Response: Dear reviewer, thank you for your valuable feedback and for your constructive comment, which has helped improve the quality of our manuscript. We appreciate your suggestion regarding the inclusion of the PRISMA Flow Diagram. In response, we have revised the manuscript to incorporate the PRISMA Flow Diagram, which now provides a clearer overview of the study selection process.
Sincerely,
Dr. Milica Vasiljevic
Faculty of Medical Sciences
University of Kragujevac, Serbia
Reviewer 2 Report
Comments and Suggestions for Authors
Dear Authors;
This review about mucosal melanomas should be useful because there aren’t any recent reviews on this topic.
The study emphasizes the difficulties of diagnosing and managing sinonasal melanoma and highlights the absence of efficacy target therapies.
It investigates epidemiological, etiopathogenetic, clinical molecular, and therapeutical aspects of this tumor but histological features are not fairly mentioned. Therefore, should be better addressed.
The text is well-written, comprehensible and the English is good.
.
Author Response
Comment: The study emphasizes the difficulties of diagnosing and managing sinonasal melanoma and highlights the absence of efficacy target therapies. It investigates epidemiological, etiopathogenetic, clinical molecular, and therapeutical aspects of this tumor but histological features are not fairly mentioned. Therefore, should be better addressed.
Response: Dear reviewer, thank you for your thoughtful and constructive feedback. We appreciate your suggestion to address the histological features of sinonasal mucosal melanoma more thoroughly. In response, we have revised the manuscript to include a detailed section on the histological characteristics of this tumor: Mucosal melanomas exhibit a variety of histological patterns, making them challenging to identify. These tumors may show spindle-shaped, epithelioid, or mixed cell morphologies, with spindle cells being elongated with eosinophilic cytoplasm, epithelioid cells being larger and polygonal with abundant cytoplasm, and mixed types containing both features. Sinonasal melanomas share common characteristics with other mucosal melanomas but have site-specific differences, including an infiltrative growth pattern, high mitotic activity, and frequent necrosis and ulceration. Vascular invasion is common, which contributes to their metastatic potential. Immunohistochemical staining, revealing markers like S-100, HMB-45, and Melan-A, plays a crucial role in diagnosis, while negative markers like cytokeratins help distinguish them from carcinomas. Differential diagnosis includes undifferentiated carcinomas, lymphomas, olfactory neuroblastomas, and sarcomas. Histologically, mucosal melanomas often lack pigmentation, complicating diagnosis, and may show high mitotic rates linked to worse outcomes. Given the wide range of morphological features, these melanomas can overlap with various sinonasal tumors, making immunohistochemical evaluation essential for accurate classification [53,67,72-75].
We believe this addition provides a more comprehensive understanding of the topic and enhances the overall quality of the review.
Thank you again for your valuable input, which has helped improve our manuscript.
Sincerely,
Dr. Milica Vasiljevic
Faculty of Medical Sciences
University of Kragujevac, Serbia
Reviewer 3 Report
Comments and Suggestions for Authors
Nasal and paranasal sinus mucosal melanoma: Long-term survival
outcomes and prognostic factors (https://doi.org/10.1016/j.amjoto.2021.103070) confirms the survival percentage of MMs is around 25% compared to cutaneous melanoma (around 80%).
Colleen et al. In his study he states that among non-cutaneous melanomas the paranasal sinuses are the second in incidence compared to non-cutaneous melanomas. Furthermore, among non-cutaneous melanomas he states that the most frequent in terms of incidence are the anorectal ones.
I also found this article interesting (https://doi.org/10.3390/jcm8101577) which talks about molecular pathways in mucosal melanomas.
I must also say that it is one of the few reviews that includes such a high number of studies
Author Response
Comment 1: Nasal and paranasal sinus mucosal melanoma: Long-term survival outcomes and prognostic factors (https://doi.org/10.1016/j.amjoto.2021.103070) confirms the survival percentage of MMs is around 25% compared to cutaneous melanoma (around 80%).
Response 1: Dear reviewer, thank you for your insightful comment. We appreciate your suggestion to include survival data for mucosal melanomas in comparison to cutaneous melanomas. In response, we have added the following sentence to the main text: "Patients with MM, including sinonasal melanoma, typically have a poorer life expectancy compared to those with cutaneous melanoma [42]. Abt and colleagues [104] reported that the survival rate for mucosal melanomas is approximately 25%, compared to 80% for cutaneous melanomas."
Comment 2: Colleen et al. In his study he states that among non-cutaneous melanomas the paranasal sinuses are the second in incidence compared to non-cutaneous melanomas. Furthermore, among non-cutaneous melanomas he states that the most frequent in terms of incidence are the anorectal ones.
Response 2: In response, we have added the following sentence regarding the incidence of non-cutaneous melanomas to the main text: "In a study conducted by Colleen McLaughlin and colleagues [35], the paranasal sinuses are the second most common site for non-cutaneous melanomas, following anorectal melanomas, which have the highest incidence among these types."
Comment 3: I also found this article interesting (https://doi.org/10.3390/jcm8101577) which talks about molecular pathways in mucosal melanomas.
Response 3: Thank you for your valuable comment and for suggesting this interesting article. In response, we have incorporated the mentioned reference into the manuscript, highlighting BRAF and other gene mutations associated with sinonasal melanoma.
Thank you again for your constructive feedback.
Sincerely,
Dr. Milica Vasiljevic
Faculty of Medical Sciences
University of Kragujevac, Serbia